# Risk Factors and Prevalence of Abdominal Obesity among Upper-Secondary Students

**DOI:** 10.3390/ijerph16101750

**Published:** 2019-05-17

**Authors:** Ewa Błaszczyk-Bębenek, Beata Piórecka, Małgorzata Płonka, Izabela Chmiel, Paweł Jagielski, Katarzyna Tuleja, Małgorzata Schlegel-Zawadzka

**Affiliations:** 1Human Nutrition Department, Institute of Public Health, Faculty of Health Sciences, Jagiellonian University Medical College, Grzegórzecka 20, 31-531 Krakow, Poland; beata.piorecka@uj.edu.pl (B.P.); paweljan.jagielski@uj.edu.pl (P.J.); tulejakatarzyna@gmail.com (K.T.); m.schlegel-zawadzka@uj.edu.pl (M.S.-Z.); 2Department of Anatomy, University of Physical Education in Krakow, al. Jana Pawła II 78, 31-571 Krakow, Poland; malgorzata.plonka@gmail.com; 3Department of Medical and Environmental Nursing, Institute of Nursing and Midwifery, Faculty of Health Sciences, Jagiellonian University Medical College, Michałowskiego 12, 31-126 Krakow, Poland; izabela.chmiel@uj.edu.pl

**Keywords:** waist circumference, abdominal obesity, WHtR, WHR, adolescence

## Abstract

Inadequate eating habits, as well as a low level of physical activity, influence adipose tissue deposition. The aim of the study was to assess the prevalence of central obesity in upper-secondary students and to determine the factors related to its occurrence. The survey included 309 participants, aged 16 to 18 years from Krakow (Poland). Anthropometric measurements were taken during the periodic assessment of students’ health status. An anonymous questionnaire was used to assess the nutritional and non-nutritional risk factors of participants. According to different methods of measurement, abdominal obesity (AO) was observed in 15.5% (WC—waist circumference), 10.7% (WHtR—waist to height ratio) or 21.7% (WHR—waist to hip ratio) participants. Abdominal obesity (WC) was significantly associated with family history of excess body weight and higher economic status of the family. The risk of AO (WC) was significantly lower among adolescents who declared higher physical activity. Boys who eat first breakfasts have lower AO risk according to WHtR interpretation. Abdominal obesity in gender group was related to the self-esteem of one’s own appearance according to WHtR and WC. Abdominal obesity was associated with the family environment and modifiable lifestyle factors and was dependent on gender.

## 1. Introduction

Obesity, being a condition of having excessive adipose tissue in the body, may lead to and exacerbate a number of health problems [1,2,3]. Although the ratio most frequently used for a body mass assessment, including obesity, is BMI (body mass index), other indices linked to abdominal obesity may predict the risk of some chronic diseases in a more precise way [4]. Adipose tissue, located in the abdominal cavity, constitutes a risk factor of various diseases of metabolic origin [5] and it may be better assessed by WC (waist circumference), WHtR (waist-to-height ratio) or WHR (waist-to-hip ratio). Since WC does not take into account differences in height and thus may under- or over-evaluate the risks for short and tall individuals, respectively, WHtR was proposed as more reliable in that respect. As for WHR, it is more gender-sensitive because the typical body structure differs for men and women—android (apple shaped) vs. gynoid (pear shaped) type, respectively [6]. 

According to research, the problem of excessive adipose tissue concerns not only adults but also children and adolescents [1,2,3]. Regardless of the definition, an association of abdominal obesity in children and adolescents with cardio-metabolic risk factors was observed [7]. Abdominal obesity is also associated with more risks of metabolic syndrome in children and adolescents [8,9]. In the light of a Spanish study, a group of children and adolescents with normal weight (based on BMI) had an increased degree of abdominal obesity. This result highlights the importance of waist circumference measurement in the screening of young people’s health status [10]. Worldwide data from research of children and adolescents show that abdominal obesity is an important public health problem. In a Turkish adolescent population abdominal obesity was prevalent in overweight group according to WC and WHtR, as well as in the obese groups [11]. Anthropometric data of Brazilian schoolchildren from the period of 2000–2015 showed that WC and the prevalence of abdominal obesity significantly increased, regardless of gender [12]. Central obesity (WHtR ≥ 0.5) significantly increased in a 30 year period of time in five cross-sectional, children and adolescent population surveys conducted in Australia [13]. Data from four cross-sectional surveys, conducted in Poland between 1966 and 2012, also showed a tendency to a greater increase of central obesity than the overall one among children and adolescents. Abdominal fat deposit leads to higher health risks so in the future we are likely to observe a greater number of metabolic complications in Polish children and adolescents [14]. Schröder et al. proposed that abdominal obesity identification by WC needs to be included in a routine clinical practice [10]. This was also the reason for conducting our research and investigating the aspect of a nutritional status of children and adolescents.

The etiopathogenesis of central obesity is multifactorial [15,16,17,18,19,20]. Therefore, the purpose of this paper was to assess the prevalence of central obesity in upper-secondary students as well as evaluate the impact of selected factors, including family history of obesity and economic status, physical activity, selected eating habits and self-assessment of one’s own appearance on abdominal obesity in adolescents.

## 2. Materials and Methods 

The study was designed as a cross-sectional survey, comprising 309 students (141 boys and 168 girls) aged 16–18 years and sporadically older (by a few months) who attended five upper-secondary schools in Krakow (Southern Poland). Participation in the study was voluntary and anonymous (the response rate was about 35%). All students aged between 16 and 18 years were invited. Informed consent was signed by study subjects or their legal guardians. This study was reported according to the Strengthening the Reporting of Observational Studies in Nutritional Epidemiology (STROBE-nut) checklist [21]. 

The study was conducted according to the ethical principles for medical research stated in the Helsinki Declaration [22]. The study was carried out in 2011 in selected schools from Krakow equipped with a school nurse’s room, to ensure privacy. The anthropometric measurements (height, body mass, waist and hip circumference) were performed by the students of nursing and under the supervision of the school nurse and an internship supervisor. Height and body mass measurements are part of a screening test of student’s health status, obligatory in Polish schools [Minister of Health regulation of 28 August 2009 concerning organisation of preventive medical care for children and adolescents (Dz. U. of 2009 Nr. 139, poz. 1133)] [23]. Since waist and hip circumference measurements are not obligatory, they required consent, just like an anonymous questionnaire about risk factors. 

### 2.1. Anthropometric Assessment

The adolescents’ anthropometric data comprised body mass measured with medical weight (standardised to 0.1 kg) without shoes and in light clothes, and body height was measured in Frankfurt Plane position with a height meter within an accuracy of 0.1 cm. The waist circumference (WC) measurement was taken in a standing position, with a tape measure within an accuracy of 0.1 cm, midway between the rib and the iliac crest. The BMI value was calculated (according to the formula body mass (kg)/height (m^2^)), for adolescents younger than 18 years following the OLAF project (growth references for Polish school-aged children and adolescents); the value exceeding the 95th percentile was classified as obesity and the value between the 85th and 95th percentile as overweight [24]. Adult persons were classified according to the BMI classification for adult people [25]. 

Abdominal obesity was assessed according to WC ratio as well as on the basis of growth charts, which were developed in the OLAF project in which it was identified with a ratio of 95th percentile for both genders [26]. In the case of adult subjects, WC value was calculated as ≥94 cm for boys and ≥80 cm for girls, taken as a criterion of abdominal obesity prevalence [27]. While evaluating abdominal obesity, WHtR was also used, with the value ≥0.5 indicating the prevalence of obesity of this type [9]. WHtR was calculated according to the formula waist circumference (waist (cm)/height (cm). However, for the purpose of WHR (according to the formula waist circumference (cm)/hip circumference (cm)), border points’ value was applied according to WHO for adolescents above 16 years old. In the case of the value ≥0.9 in boys and ≥0.85 in girls, the android-type obesity was observed [28]. 

### 2.2. Questionnaire

The anonymous survey concerned above all personal, socioeconomic and lifestyle data. It also included age in completed years, gender, place of residence (rural, urban populations) and educational level according to age. Subjects were classified into two levels of physical activity based on two questions, one about the frequency of physical activity (everyday; four to five times per week; two to three times per week; one time per week and one time per month), and the second about time spent in physical exercise (≥4 h; 2–3 h; 1–2 h; 30 min to 1 h; ≤30 min). The level of physical activity was based on Polish guidelines; low level—less than 1 h physical exercise/7 times per week, higher level ≥1 h 7 times per week [29]. In the survey, the subjects were also asked about their body self-assessment (satisfaction with own body weight: no and yes answer).

The subjects answered questions about the family—number of persons in household, siblings, parents’ work activity, educational level of parents (according to type of education—primary, vocational, secondary, higher), family history related to the prevalence of excessive body weight (incidence of overweight or obesity among parents or grandparents: no vs. sum of responses yes and I don’t know) as well as family economic status (question: do your parents always have enough money to buy the food they want to buy?—in four-level scale of answers: yes, sometimes yes, I don't know and no). 

The food frequency questionnaire (FFQ) used in the study was applied to assess eating habits. This questionnaire, which concerned the frequency of consumption of selected products (26 items from different groups) and beverages (8 items), was similar to a validated Polish questionnaire prepared by the Behavioral Conditions of Nutrition Team, Committee of Human Nutrition Science, Polish Academy of Sciences [30]. The frequency of consumption was assessed on the scale—daily, weekly or monthly. The survey did not include open-ended questions. A 6-point scale was used to assess the frequency of consuming selected products, food and beverages.

### 2.3. Data Analysis

All the collected data were analysed statistically with the use of the IMAGO PRO Academic 5 software pack—IBM SPSS Statistics 25 (Predictive Solutions, Cracow, Poland). A chi-square test was used to compare the differences in categorical variables. In order to examine differences depending on gender, the U Mann–Whitney test was used however, between age groups the Kruskal–Wallis test or one-way analysis of variance (ANOVA) were used taking in a significance level of α = 0.05. Odds ratios between selected risk factors (family history of obesity and economic status, physical activity, selected eating habits and self-assessment of body mass) for abdominal obesity (WC and WHtR as the dependent variables) in total and gender groups were calculated using binary logistic regression (95% confidence intervals, CI). 

## 3. Results

There were 309 students participating in the survey, 141 (46%) boys and 168 (54%) girls. The average age of the students was 17.20 ± 0.91 years (17.01 ± 0.84 for boys and 17.35 ± 0.94 years for girls, *p* = 0.0007). The sixteen-year-old students constituted the biggest group (44.7%), 29.1% of students were seventeen years old and a group of 26.2% comprised eighteen-year-old students. Adolescents, as total group, were high school students. Detailed socio-economic and family characteristics of the participants are presented with reference to gender and age in Table 1. There were no significant differences in socio-economic and family characteristics in relation to gender. However, differences in the education of mothers (*p* = 0.0015) and fathers (*p* = 0.0063) were found depending on age groups. 

BMI interpretation with reference to gender and age is shown in Table 2. There were no statistically significant differences in the interpretation of BMI between groups of boys and girls (*p* = 0.1530). There were also no statistically significant differences in the interpretation of BMI between age groups (*p* = 0.0913). 

### 3.1. Anthropometric Characteristics and Abdominal Obesity Prevalence

Anthropometric measurements included selected parameters and indicators of abdominal obesity of adolescents with reference to gender and age, as shown in Table 3. Characteristics of abdominal obesity indicators including gender in age groups are shown in Table 4.

Waist circumferences of boys and girls were significantly different (*p* = 0.0001). Taking WC into consideration, android-type obesity occurred in 15.5% of the respondents. A major difference was observed in WC interpretation concerning the age of the group examined (*p* = 0.0123). Abdominal obesity was prevalent more often in 18 year old students (23.5%), whereas in the group of 16 year olds it concerned 13.8% of students, and in the group of 17 year olds (11.1%) it occurred least frequently. 

The WHtR indicated the prevalence of abdominal obesity in 10.7% of all the examined students. However, there were no statistically significant differences in the prevalence of abdominal obesity depending on gender and age (boys—12.8% and girls—8.9%). Abdominal obesity according to WHtR index concerned 10.9% of 16 year olds and it was similar in the groups of 17 year olds (11.1%) and 18 year old students (9.9%). In the last group, abdominal obesity was more common in boys than in girls (18.2% vs. 5.8%; *p* = 0.0370). 

WHR interpretation showed that abdominal obesity was present in 21.7% of respondents and it was significantly higher in boys (33.3%) than in girls (11.9%; *p* < 0.0001). Taking age groups into consideration, abdominal obesity occurred in 31.2% of 16 year olds, in 20% of 17 year olds and in 7.4% of 18 year olds according to WHR. 

### 3.2. The Family History of Overweight and Obesity

Overall, 25.6% of adolescents declared overweight and obesity in family history. In the total group, abdominal obesity assessed by WC (≥95th percentile) was associated with prevalence of overweight and obesity in parents or grandparents (no vs. sum of responses yes and sometimes yes) of the subjects [OR = 3.09; 95% CI (1.574–6.052)]. The odds of abdominal obesity (WC) in the girls’ group were 2.56 times higher for those with obesity in family history [95% CI (1.099–5.968)]. In the boys’ group, both adiposity indicators WC (≥95th percentile) and WHtR were 3.76 times more often associated with prevalence of overweight and obesity in parents or grandparents [95% CI (1.174–11.822)].

### 3.3. Family Economic Status

Taking into account the economic situation of the families, 85.1% of them could buy sufficient food. Only 9.1% of the respondents admitted that they could not afford to buy what they wanted, and 4.2% of the respondents were not able to estimate their family financial situation in this context. Five students (1.6%) admitted that their parents sometimes could not afford to buy food.

Economic situation (yes vs. sum of response no, I don’t know and sometimes yes) of family in total group was associated with higher risk of abdominal obesity by WC (≥95th percentile). Adolescents from families in a better economic situation in comparison with those whose economic situation was worse had 9.84 times higher risk of abdominal obesity [95% CI (1.323–73.169)]. 

### 3.4. Physical Activity 

Both school and after-school activities were taken into account for estimating the level of physical activity. It was found that 11% of boys and 10% of girls did not attend physical education classes at school. Boys (30%) compared with girls (18%) more frequently practiced after-school physical activity (*p* = 0.0001). Duration time of training in the case of boys was also significantly higher (*p* = 0.0001) than in girls. Individual workouts in the group of boys lasted 120 min on average while in the group of girls it was only 30 to 60 min. Adolescents in total whose physical activity was over 1 h/7 times per week have 61% less risk of abdominal obesity according to WC [OR = 0.39; 95% CI (0.180–0.839)] especially in the girls’ group [OR = 0.20; 95% CI (0.046–0.891)]. 

### 3.5. Selected Eating Habits 

The survey did not show statistically significant relationships between the frequency of consumption of selected products and beverages and abdominal obesity among the respondents. Adolescents from Krakow usually ate four meals per day (39.5%); however, only 8.7% of the subjects ate fewer than three meals. Almost half of the adolescents (49.2%) chose sweets as their snacks between meals. The majority of the subjects declared eating dinner (94.2%), whereas afternoon tea was the most commonly skipped meal (67.3%). Also, 12.3% of adolescents did not eat the first breakfast and 26.5% did not eat the second breakfast (the meal between breakfast and lunch). Having breakfast was related to a significantly lower risk of prevalence of abdominal obesity only in the group of boys, according to WHtR [OR = 0.28; 95% CI (0.085–0.925)].

### 3.6. Self-Assessment of One’s Own Appearance

According to the survey results, girls were more often dissatisfied with their looks than boys (47% vs. 18.4%; *p* < 0.0001). Despite not being satisfied with their appearance, most of the girls (74%) considered their body mass as normal. Girls more frequently than boys (*p* < 0.0001) decided to go on a diet (30% girls and 8.5% boys). 

Additionally, the study examined how abdominal obesity affects the perception of one's own figure—adolescents with abdominal obesity based on WC had an 83% lower chance of satisfaction with their own figure in comparison to adolescents who did not have obesity [OR = 0.17; 95% CI (0.089–0.338)] and respectively 78% lower risk by WHtR [OR = 0.26; 95% CI (0.100–0.464)]. Also, in gender group, odds of lower satisfaction with own appearance were in girls with abdominal obesity by WC [OR = 0.13; 95% CI (0.046–0.356)] and WHtR [OR = 0.17; 95% CI (0.025–0.535)] as well as in boys with abdominal obesity identified by WC [OR = 0.21; 95% CI (0.075–0.616)] and WHtR [OR = 0.16; 95% CI (0.056–0.461)]. 

## 4. Discussion

Abdominal obesity was identified in 15.5% (WC), 10.7% (WHtR) and 21.7% (WHR) of the examined students. Similarly, in a Turkish adolescent girls’ population (12–18 years), abdominal obesity (WC ≥ 90th percentile and WHtR ≥ 0.5) was found in 16.9% (WC) and 10.4% (WHtR) of subjects [11]. In the light of the results, abdominal obesity assessed according to WC was more frequent in the oldest age group (18 years and older)—23.5%. Carrying out surveys among adolescents from Wrocław, Mikołajczak et al. [31] assessed metabolic syndrome risk factors, including WC—like in our study the results were significantly different in boys (79.0 ± 5.0 cm) than in girls (70.0 ± 4.5 cm). The waist circumference of ≥90th percentile occurred in 17.6% of boys and 22.2% of girls. Abdominal obesity concerned in total 20% of the examined adolescents (778 in total) in the same age group [31] and was similar as in our studies. According to the research of Broniecka et al. [32], abdominal obesity evaluated on the basis of WC above the 75th percentile occurred in 45% of 17 to 18 year old adolescents, attending upper-secondary schools in Wrocław. However, Piotrowska et al. [33] diagnosed abdominal obesity in 20.6% of adolescents from Wrocław, according to WC ≥ 95th percentile like in our research. Taking into account data of Klimek-Piotrowska et al. [34], WC was significantly higher in boys (75.8 ± 9.1 cm) than in girls (69.6 ± 7.5 cm), for 14 to 18 year old students from Krakow, which was confirmed in our results, too. Being female, inter alia, was associated with an increased risk of abdominal obesity in a 15 year and older group of adolescents from China [35].

Waist to height ratio as a good indicator was used in much research, inter alia, in a large European cross-sectional survey of children and adolescents from Łódź (Poland) [34,36]. According to studies carried out among Greek adolescents, WHtR was also found as a good indicator for both genders [37]. However, the results from a systematic literature review suggest that the WHtR cut-off point for children and adolescents aged 6 to 18 years should be lower than that determined for adults. The weighted average cut-off points, based on the data from seven studies, were 0.459 (± 0.017) for girls and 0.473 (± 0.019) for boys [38]. 

According to the NHANES study (National Health and Nutrition Examination Survey), abdominal obesity (WHtR) was observed in 32.9% of children and adolescents aged 6 to 18 years [39], which implies that it was prevalent 3 times more often than in our study (10.7%). Also compared to our results, Klimek-Piotrowska et al. [34] observed a significant difference in the prevalence of abdominal obesity in girls and boys on the basis of WHtR which not occur in our own study [34].

In our survey from Krakow, the declared prevalence of overweight and obesity in the participants’ families was connected with overweight and obesity (in reference to BMI) in both genders, as well as with prevalence of central obesity. Piotrowska et al. [33] stated that abdominal obesity more frequently concerned girls from families with obesity or obesity together with diabetes occurrences. The family factors, related to the prevalence of obesity, were stressed by Suder et al. and Melzer et al. [40,41]. In the CASPIAN-V study, significant correlations were observed between waist circumferences of parents and their children. The risk of abdominal obesity in a multivariate model was also in those cited research where examined parents had excess body weight [15]. Abdominal obesity in German primary school children was associated with parental obesity [OR 1.95; 95% CI (1.22–3.10)] as well as a higher educational level [OR 0.64; 95% CI (0.42–0.98)] [16]. The prevalence of overweight/obesity was found to be higher among European families with overweight/obese parents [42]. 

Socioeconomic status and maternal occupation level were associated with abdominal obesity prevalence (WHtR) in European girls (HELENA-CSS) [18]. Similar findings were demonstrated by Castro et al. where the mother’s lower education level was associated with abdominal obesity, too [19]. In our study it was determined that family situation of the students who attended upper-secondary schools was related to abdominal obesity prevalence. WC (≥95th percentile) was higher in our adolescents’ group with better home economic situation [OR = 9.837; 95% CI (1.323–73.169)]. Długosz et al. [43] presented different findings where high socio-economic status of the examined families was associated with a low risk of abdominal obesity prevalence, since it concerned only 3% of the subjects from mostly rural areas. 

According to our research, a higher physical activity (over 1 h/7 times per week) reduces the risk of abdominal obesity (WC) by 61% in the total study group. Following Suder et al. [40], low physical activity was a major risk factor of abdominal obesity prevalence in boys [OR 1.91; 95% CI (1.06–3.43)]. A decline in adipose, visceral and subcutaneous tissues was observed on the basis of IRAS Family Study (Insulin Resistance Atherosclerosis Study)research in subjects practising exercises compared to those who practised rarely or not at all [44]. Factors such as age and inadequate physical activity levels and increased screen time were associated with higher odds of total and central obesity in Greek children from the National Action for Children’s Health (EYZHN) program [45]. A relationship between unhealthy lifestyle (e.g., physical inactivity) and abdominal obesity in a cross-sectional study among Brazilian adolescents, which involved 62,063 students (12–17 years) was confirmed [46]. 

With reference to our research, adolescents with abdominal obesity had an 83% lower chance of satisfaction with their own figure in comparison to adolescents who did not have obesity based on WC [OR = 0.17; 95% CI (0.089–0.338)] and a 78% lower risk by WHtR [OR = 0.26; 95% CI (0.100–0.464)]. Our research demonstrated that girls less frequently accepted their body and were more critical about their appearance. As noted by Piotrowska et al. [33], self-assessment of their appearance, including body weight, is an important factor influencing eating habits of young people, which has been reflected in this paper. 

Irregular breakfasts and meals are related significantly to an increased risk of abdominal obesity prevalence [17,47,48]. According to research carried out in a group of girls from rural areas, irregular breakfasts and skipping meals throughout the day were significantly related to the prevalence of abdominal obesity [39]. Despite the fact that our study did not show a correlation between the consumption of selected products and the parameters of abdominal obesity, other researchers emphasised such a relationship [49].

However, our study had several limitations. First of all, the size of the examined group and response rate (35%). Second, only selected anthropometric indicators were taken into account in the study. The prevalence of abdominal obesity was not assessed with bioelectrical impedance analysis (BIA) or dual-energy X-ray absorptiometry (DXA), which would give a more precise result. Third, there were also limitations in the evaluation of dietary habits that influence the outcome of the study. Only a frequency questionnaire of consumption of selected products and beverages was used, which may have contributed to the lack of association between dietary factors and abdominal obesity incidence.

## 5. Conclusions

In the study, the prevalence of abdominal obesity was identified in 10.7–21.7% of examined adolescents using different indicators. The family history related to excessive body mass, socioeconomic status and low physical activity played a vital role in the prevalence of central obesity. The relation between skipping breakfast and central obesity prevalence has been confirmed. Modifiable risk factors of abdominal obesity in adolescents should be taken into consideration and appropriate preventive actions should be introduced, including the development of proper eating habits primarily among boys and encouraging primarily girls to practice physical activity.

## Figures and Tables

**Table 1 ijerph-16-01750-t001:** Socio-economic and family characteristics in total and with reference to gender and age group [%].

Parameter	In Total (309)	Boys (141)	Girls (168)	16 Years (138)	17 Years (90)	18 Years (81)
Place of residence						
rural	34.2	37.6	31.3	27.9	34.4	44.4
urban	65.8	62.4	68.7	72.1	65.6	55.6
Level of education						
1	38.2	42.6	34.5	83.3	3.3	0
2	32.0	46.1	20.3	16.7	82.3	2.5
3	29.8	11.3	45.2	0	14.4	97.5
Number of persons in household						
≤3	23.0	21.3	24.7	27.0	18.9	21.0
4	37.5	36.2	39.2	37.5	34.4	42.0
5≥	39.5	42.5	36.1	35.5	46.7	37.0
Siblings						
no	14.9	16.3	13.7	14.5	13.3	17.3
yes	85.1	83.7	86.3	85.5	86.7	82.7
Mother education *						
primary	1.0	0.7	1.2	1.5	0	1.2
vocational	16.1	17.6	15.0	22.2	6.8	16.0
secondary	43.4	47.4	40.1	43.0	40.9	46.9
higher	39.5	34.3	43.7	33.3	52.3	35.9
Father education *						
primary	0.7	1.5	0	1.5	0	0
vocational	26.2	33.1	20.5	30.4	17.2	28.7
secondary	37.4	32.4	41.5	34.1	34.5	46.3
higher	35.7	33.0	38.0	34.0	48.3	25.0
Mother work activity						
No	16.8	15.6	17.9	14.5	15.6	22.2
Yes	83.2	84.4	82.1	85.5	84.4	77.8
Father work activity						
No	10.7	8.0	12.9	12.1	10.1	8.9
Yes	89.3	92.0	87.1	87.9	89.9	91.1

*—statistically significant difference between age groups, Kruskal–Wallis test.

**Table 2 ijerph-16-01750-t002:** Body mass index (BMI) interpretation with reference to gender and age [%].

Study Group (*n*)	Underweight	Risk of Underweight	Standard	Overweight	Obesity
In total (309)	2.3	7.0	77.7	9.4	3.6
Boys (141)	0.7	4.3	82.3	9.2	3.5
Girls (168)	3.6	9.5	73.8	9.5	3.6
16 years (138)	2.2	2.9	81.9	8.7	4.3
17 years (90)	0	8.9	76.7	11.1	3.3
18 years (81)	4.9	12.3	71.6	8.6	2.5

*n*—number of students.

**Table 3 ijerph-16-01750-t003:** Mean values (X ± SD) of anthropometric measurements in total and with reference to gender and age.

Parameter	In total (*n* = 309) X ± SD	Boys (*n* = 141) X ± SD	Girls (*n* = 168) X ± SD	16 years (*n* = 138) X ± SD	17 years (*n* = 90) X ± SD	18 years (*n* = 81) X ± SD
height [cm] *	172.75 ± 8.36	179.01 ± 5.98	167.50 ± 6.18	174.00 ± 8.24	175.17 ± 7.83	170.01 ± 8.17
body weight [kg]	63.33 ± 10.42	69.44 ± 9.29	58.21 ± 8.37	63.50 ± 11.12	65.54 ± 9.94	60.95 ± 9.85
hip circumference [cm] *	92.97 ± 8.08	91.98 ± 8.86	93.81 ± 7.27	91.00 ± 7.83	92.93 ± 8.75	95.04 ± 7.06
waist circumference [cm] *	76.27 ± 8.66	79.43 ± 8.64	73.61 ± 7.76	76.00 ± 9.77	77.09 ± 8.11	74.81 ± 8.07
WHtR [cm/cm]	0.44 ± 0.05	0.44 ± 0.05	0.44 ± 0.05	0.43 ± 0.06	0.44 ± 0.05	0.44 ± 0.04
WHR [cm/cm]	0.82 ± 0.09	0.87 ± 0.09	0.79 ± 0.07	0.85 ± 0.09	0.82 ± 0.09	0.78 ± 0.06

*n*—number of respondents, X—average, SD—standard deviation, WHtR—waist to height ratio, WHR— waist to hip ratio, *—statistically significant difference between gender groups, U Mann–Whitney test.

**Table 4 ijerph-16-01750-t004:** The average height, body weight and fatness indexes values with reference to gender in age group of the examined students.

Parameter/Adiposity Indicators	16 Years Old *n* = 138)		17 Years Old (*n* = 90)		18 Years Old and Older (*n* = 81)	
	Boys X ± SD	Girls X ± SD	Boys X ± SD	Girls X ± SD	Boys X ± SD	Girls X ± SD
height (cm)	179.13 ± 8.24	168.26 ± 6.72	179.32 ± 6.06	168.11 ± 4.89	178.25 ± 6.39	166.84 ± 6.38
body weight (kg)	64.03 ± 11.12	57.64 ± 8.76	68.83 ± 9.13	59.93 ± 8.75	69.25 ± 9.52	57.76 ± 7.98
hip circumference (cm) *, **	90.29 ± 7.83	90.89 ± 7.54	92.61 ± 9.48	93.47 ± 7.44	93.88 ± 8.15	95.49 ± 6.60
waist circumference (cm)	79.53 ± 10.00	75.04 ± 8.95	79.02 ± 7.62	73.81 ± 7.97	80.09 ± 8.51	72.08 ± 6.60
WHtR (cm/cm)	0.44 ± 0.06	0.45 ± 0.06	0.44 ± 0.04	0.44 ± 0.05	0.45 ± 0.05	0.44 ± 0.04
WHR (cm/cm) *	0.88 ± 0.09	0.81 ± 0.07	0.86 ± 0.09	0.76 ± 0.05	0.84 ± 0.06	0.76 ± 0.05

*n*—the number of objects, X—average, SD—standard deviation, *—statistically significant difference between age groups in girls, ANOVA test, **—statistically significant difference between age groups in boys, ANOVA test.

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
