# Peer review of "Risk Factors and Prevalence of Abdominal Obesity among Upper-Secondary Students"

_ijerph, 2019, doi:10.3390/ijerph16101750_

Round 1
Reviewer 1 Report
Reviewer’s Comments Manuscript ID: Ijerph-488446
The paper presents findings from a seemingly interesting study, but authors need to do major revisions/editing and perhaps re-submit the paper to be considered for a second review.
The paper has several weaknesses. Some of them include the following:
1) The abstract should to be revised. It needs editing. Some of the sentences are not clear. For example, the lines 22 and 23…authors write “An anonymous questionnaire was a research tool with questions concerning the nutritional and non-nutritional risk factors’ This is sentence is a bit awkward, authors can eliminate words that are not necessary, and make the sentence too long.… So, authors can say, for instance, ‘An anonymous questionnaire was used to assess the nutritional and non-nutritional risk factors of participants.’
2) Authors should consider defining the following abbreviations: “WC”, “WHtR”, “WHR” in the abstract.
3) I think the abstract is complete. Authors should consider including concluding statement in the abstract.
4) The introduction also needs to be revised, it is quite weak. Some of the sentences are incoherent, sentences do not flow together. This makes it a bit difficult for the reader to follow the ideas that authors are trying to communicate.
5) The description of the study methods and materials looks a bit okay, but again some of the sentences are awkward. For example, the sentence that begins from line 79. “The inclusion criteria in the study was the age > 16 years, informed consent to participate in the study of the study the subjects and their legal guardians, and the possibility to perform anthropometric measurements in the study subject.” What are authors trying to communicate in this sentence? This sentence is not clear. Another example of such sentences is the one on lines 91 and 94, and other sections in the paper.
6) The presentation of the findings also needs to be edited and condensed. Who are “conducted participants”? Please see line 155… Are you referring to the study participants? The sentence on lines 239 and 240, “The first breakfast did not eat 12.3% adolescents and 26.5% of them did not eat the second breakfast too.” What does this sentence mean? What are authors trying to say in this sentence?
The discussion is weak. Please discuss the major findings. You don’t discuss or repeat all your findings in the discussion. Be selective of how you support your major findings with other works or literature. Please condense the discussion.
Author Response
Response to Reviewer 1 Comments
Point 1: The abstract should to be revised. It needs editing. Some of the sentences are not clear. For example, the lines 22 and 23…authors write “An anonymous questionnaire was a research tool with questions concerning the nutritional and non-nutritional risk factors’ This is sentence is a bit awkward, authors can eliminate words that are not necessary, and make the sentence too long.… So, authors can say, for instance, ‘An anonymous questionnaire was used to assess the nutritional and non-nutritional risk factors of participants.
Response 1: Changes in the text were made – line 22-25 and 29-31.
Point 2: Authors should consider defining the following abbreviations: “WC”, “WHtR”, “WHR” in the abstract.
Response 2: Changes in the text were made – line 24-25
Point 3: I think the abstract is complete. Authors should consider including concluding statement in the abstract.
Response 3: Changes in the last sentence was made – line 30-31.
Point 4: The introduction also needs to be revised, it is quite weak. Some of the sentences are incoherent, sentences do not flow together. This makes it a bit difficult for the reader to follow the ideas that authors are trying to communicate.
Response 4: Changes in the text were made.
Point 5: The description of the study methods and materials looks a bit okay, but again some of the sentences are awkward. For example, the sentence that begins from line 79. “The inclusion criteria in the study was the age > 16 years, informed consent to participate in the study of the study the subjects and their legal guardians, and the possibility to perform anthropometric measurements in the study subject.” What are authors trying to communicate in this sentence? This sentence is not clear. Another example of such sentences is the one on lines 91 and 94, and other sections in the paper
Response 5: Duplicate information has been deleted
Point 6: The presentation of the findings also needs to be edited and condensed. Who are “conducted participants”? Please see line 155… Are you referring to the study participants? The sentence on lines 239 and 240, “The first breakfast did not eat 12.3% adolescents and 26.5% of them did not eat the second breakfast too.” What does this sentence mean? What are authors trying to say in this sentence?
Response 6: Changes in the text were made (line 155 and 232-233).
Point 7: The discussion is weak. Please discuss the major findings. You don’t discuss or repeat all your findings in the discussion. Be selective of how you support your major findings with other works or literature. Please condense the discussion.
Response 7: Some changes in the text were made.
Reviewer 2 Report
The English is not clear enough to be able to comprehensively review this study. The writing is ambiguous in places which makes it difficult to understand what the authors mean.
For example:
Line 28: 'who did not leave their first breakfasts' - does this mean they did not eat breakfast?
32: modifiable lifestyle factors dependent gender - does this mean that the lifestyle factors associated are different for male and female?
Check spelling, the correct tense and the correct word is used.
For example:
96: examine - do you mean exam or examined or examination?
119: frequency not frequently
OLAF study - provide more information about this, why was this used for the percentiles?
Introduction: The authors describe cross-sectional studies conducted in Poland in the past, state why this current study is needed. Are there no recent cross-sectional studies? State why this study is important and how the results will inform policy or interventions.
Methods
80: Did the inclusion criteria also include a maximum age of 18?
Were all children aged 16 and over in the schools invited?
Physical activity levels: the levels of physical activity are confused by the coding number (everyday-5), why are these codes added to the text?
Family questions: remove the codes, yes/no is sufficient
FFQ - provide more information, how many questions?, which selected products and beverages? Describe first breakfast and second breakfast
Results:
How many students were invited to participate? What was the response rate?
Table 2: Nutritional status indicates that there are a range of measures - use the term BMI or body size.
180: Why is the WC of boys compared to girls? Wouldn't it be expected to be different? Remove this sentence as android-type obesity is described so the WC comparison does not add anything.
3.3 This paragraph is poorly written so it is difficult to understand the results. Clearly state what the result means. A family history of obesity was associated with obesity in girls (p = x) but not in boys (p= x).
3.3 - Numbers are repeated
3.3 Family economic status: The original question was about whether the family could afford to buy food so continue to use this phrase in the results rather than the family budget was insufficient, the phrases have slightly different meanings.
3.5: Correct wording - breakfast does not eat the student, the student eats breakfast
3.6: Was the question on appearance related to body size or appearance overall (which could include hair, face etc)
Discussion
295: Instead of the term 'abnormal' state overweight/obese, do you include underweight in the definition of abnormal?
339: 'selected products' - what are these?
Limitations: was the group large enough to be able to analyse the age groups separately? Particularly in the 18 year old group? It would be simpler to report only girls and boys. What does the analysis of the age groups separately add to the study results? The response rate is not discussed, was this a limitation? Is this a representative sample?
The conclusion briefly mentions some appropriate preventive actions. Was part of the purpose of the study to inform preventive actions? If so this needs to be mentioned in the introduction and discussed in the discussion. It is important to state the implications of this study.
Author Response
Response to Reviewer 2 Comments
Point 1: Line 28: 'who did not leave their first breakfasts' - does this mean they did not eat breakfast?
Response 1: Changes in the text were made – line 28
Point 2: 32: modifiable lifestyle factors dependent gender - does this mean that the lifestyle factors associated are different for male and female?
Response 2: Changes in the text were made – line 31
Point 3: 96: examine - do you mean exam or examined or examination?
Response 3: Changes in the text were made – line 97
Point 4: 119: frequency not frequently
Response 4: Changes in the text were made – line 120
Point 5: OLAF study - provide more information about this, why was this used for the percentiles?
Response 5: Growth references for Polish school-aged children and adolescents was used. The presented BMI and waist circumference references from OLAF project are based on a current, nationally representative sample of Polish children and adolescents without known disorders affecting growth.
Changes in the text were made – line 102-104
Point 6: Introduction: The authors describe cross-sectional studies conducted in Poland in the past, state why this current study is needed. Are there no recent cross-sectional studies? State why this study is important and how the results will inform policy or interventions.
Response 6: Unfortunately, there are still few surveys on the prevalence of abdominal obesity among children and adolescents in Poland. Assessment of the prevalence of abdominal obesity is not a part of mandatory prophylactic examinations among schoolchildren, therefore such examinations as those cited and ours are an important factor that can influence decision-makers in the public health policy.
Point 7: 80: Did the inclusion criteria also include a maximum age of 18?
Response 7: We did not exclude a few students who were over 18 years of age at the time of the study (they were not over 19 years of age, though).
Point 8: Were all children aged 16 and over in the schools invited?
Response 8: Yes, all the students who were present at the school were invited to participate in the survey.
Point 9: Physical activity levels: the levels of physical activity are confused by the coding number (everyday-5), why are these codes added to the text?
Response 9: The codes have been removed from the text.
Point 10: Family questions: remove the codes, yes/no is sufficient
Response 10: Deleted from the text.
Point 11:FFQ - provide more information, how many questions?, which selected products and beverages? Describe first breakfast and second breakfast.
Response 11: FFQ included questions about the frequency of consumption of 26 products and 8 drinks typical of the Polish diet. It was asked whether the youth in general eat the first breakfast (eaten before going to school) and the second breakfast (eaten between breakfast and lunch). Most often they are not hot meals.
Point 12: How many students were invited to participate? What was the response rate?
Response 12: The response rate was about 35%.
Point 13: Table 2: Nutritional status indicates that there are a range of measures - use the term BMI or body size.
Response 13: Changes in the text were made – line 166
Point 14: 180: Why is the WC of boys compared to girls? Wouldn't it be expected to be different? Remove this sentence as android-type obesity is described so the WC comparison does not add anything.
Response 14: Deleted from the text.
Point 15: 3.3 This paragraph is poorly written so it is difficult to understand the results. Clearly state what the result means. A family history of obesity was associated with obesity in girls (p = x) but not in boys (p= x).
Response 15: Some redundant information, which did not have a significant impact on the presented results but could make it difficult for the reader to read and interpret them, was removed.
Point 16: 3.3 - Numbers are repeated
Response 16: Changes in the text were made.
Point 17: 3.3 Family economic status: The original question was about whether the family could afford to buy food so continue to use this phrase in the results rather than the family budget was insufficient, the phrases have slightly different meanings.
Response 17: Changes in the text were made – line 209-210.
Point 18: 3.5: Correct wording - breakfast does not eat the student, the student eats breakfast
Response 18: Changes in the text were made – line 232-233.
Point 19: 3.6: Was the question on appearance related to body size or appearance overall (which could include hair, face etc)
Response 19: Question on appearance was related only to information about body size.
Point 20: 295: Instead of the term 'abnormal' state overweight/obese, do you include underweight in the definition of abnormal?
Response 20: Changes in the text were made – line 286. We do not include underweight in the definition of abnormal
Point 21: 339: 'selected products' - what are these?
Response 21: 26 products and 8 drinks typical of the Polish diet.
Point 22: Limitations: was the group large enough to be able to analyse the age groups separately? Particularly in the 18 year old group? It would be simpler to report only girls and boys. What does the analysis of the age groups separately add to the study results? The response rate is not discussed, was this a limitation? Is this a representative sample?
Response 22: The sample (35% of school population, a significant number of students in each age group) seems representative enough and the response rate is not so low as to be a serious limitation. The analysis of separate age groups does show some differences, even if (in hindsight) they do not seem essential.
Point 23: The conclusion briefly mentions some appropriate preventive actions. Was part of the purpose of the study to inform preventive actions? If so this needs to be mentioned in the introduction and discussed in the discussion. It is important to state the implications of this study.
Response 23: At this stage, we do not have specific advice for preventive actions, beyond the general statements from the conclusion.
Round 2
Reviewer 1 Report
I still don't think the introduction and discussion sections of the manuscript are strong enough.
The problem I have with this manuscript mainly has to do with the quality of the presentation. The methods and results are okay, but the introduction and the discussions should to be a bit robust. It is still quite difficult to follow the flow of some of the sentences in the introduction, especially in the last 2 paragraphs of that section.
Some sections in the discussion read like summaries of related literature. Example is the paragraph 2 on page 8, lines 268 to 280. I will recommend that the authors look for a good editor to edit the manuscript or work with an editor, especially on the introduction and discussion sections to ensure they are well written.
Author Response
Response to Reviewer 1 Comments
Point 1: The problem I have with this manuscript mainly has to do with the quality of the presentation. The methods and results are okay, but the introduction and the discussions should to be a bit robust. It is still quite difficult to follow the flow of some of the sentences in the introduction, especially in the last 2 paragraphs of that section.
Response 1: Changes in the text were made (marked with a red text) also some sentences was deleted from the text.
Point 2: Some sections in the discussion read like summaries of related literature. Example is the paragraph 2 on page 8, lines 268 to 280.
Response 2: Changes in the text were made (marked with a red text) also some sentences was deleted from the text.
Reviewer 2 Report
I have reviewed the raised version. I recommend the article is accepted if the following changes are made.
Line 59: What is 'the general one'? Do you mean BMI?
Line 70: Change need to needs
Line 112, 113. Close bracket after (cm))
Line 210: remove 'they wanted'
Line 251: Change diagnosed to identified
Line 258: Change concerned to occurred
Line 282: study by Nawarycz
311: Factors such as age
319: Demonstrated in our research
327: According to research
328: regular tea drinking
332: The response rate should be included as a study limitation
340: Reword. In the study, the prevalence of abdominal obesity was ..
Author Response
Response to Reviewer 2 Comments
Point 1: Line 59: What is 'the general one'? Do you mean BMI?
Response 1: Changes in the text were made – line 53.
Point 2: Line 70: Change need to needs
Response 2: Changes in the text were made – line 63.
Point 3: Line 112, 113. Close bracket after (cm))
Response 3: Changes in the text were made – line 105.
Point 4: Line 210: remove 'they wanted'
Response 4: Deleted from the text – line 202
Point 5: Line 251: Change diagnosed to identified
Response 5: Changes in the text were made – line 243.
Point 6: Line 258: Change concerned to occurred
Response 1: Changes in the text were made – line 250.
Point 7: Line 282: study by Nawarycz
Response 7: Deleted from the text (In response to Reviewer 1).
Point 8: 311: Factors such as age
Response 8: Changes in the text were made – line 298.
Point 9: 319: Demonstrated in our research
Response 9: Changes in the text were made – line 306.
Point 10: 327: According to research
Response 10: Changes in the text were made – line 311.
Point 11: 328: regular tea drinking
Response 11: Whole sentence deleted from the text.
Point 12: 332: The response rate should be included as a study limitation
Response 12: Changes in the text were made – line 317.
Point 13: 340: Reword. In the study, the prevalence of abdominal obesity was .
Response 13: Changes in the text were made – line 325.